# COPD, but Not Asthma, Is Associated with Worse Outcomes in COVID-19: Real-Life Data from Four Main Centers in Northwest Italy

**DOI:** 10.3390/jpm12071184

**Published:** 2022-07-20

**Authors:** Stefania Nicola, Richard Borrelli, Irene Ridolfi, Virginia Bernardi, Paolo Borrelli, Giuseppe Guida, Andrea Antonelli, Carlo Albera, Stefania Marengo, Antonio Briozzo, Claudio Norbiato, Agata Valentina Frazzetto, Marina Saad, Luca Lo Sardo, Beatrice Bacco, Silvia Gallo Cassarino, Stefano Della Mura, Diego Bagnasco, Caterina Bucca, Giovanni Rolla, Paolo Solidoro, Luisa Brussino

**Affiliations:** 1SCDU Immunologia e Allergologia, AO Ordine Mauriziano di Torino, C.so Re Umberto 109, 10128 Torino, Italy; stefania.nicola@edu.unito.it (S.N.); luca.losardo@edu.unito.it (L.L.S.); 2Department of Medical Sciences, University of Torino, C.so Dogliotti, 14, 10126 Torino, Italy; richard.borrelli@edu.unito.it (R.B.); irene.ridolfi@edu.unito.it (I.R.); virginia.bernardi@edu.unito.it (V.B.); agatavalentina.frazzetto@edu.unito.it (A.V.F.); bbacco@aslcn2.it (B.B.); sil.gcassa@gmail.com (S.G.C.); stefano.dellamura@edu.unito.it (S.D.M.); caterina.bucca@unito.it (C.B.); grolla@mauriziano.it (G.R.); 3SSD Dermatologia e Allergologia, Ospedale Beauregard, Via Vaccari, 5, 11100 Aosta, Italy; pborrelli@ausl.vda.it; 4Dipartimento di Scienze Cliniche e Biologiche, University o Torino, Regione Gonzole, 10, 10043 Orbassano, Italy; giuseppe.guida@unito.it; 5SS Allergologia e Fisiopatologia Respiratoria, ASO Santa Croce e Carle, Via Michele Coppino, 26, 12100 Cuneo, Italy; antonelli.a@ospedale.cuneo.it; 6S.C. Pneumologia U, Azienda Ospedaliero-Universitaria Città della Salute e della Scienza, Corso Bramante, 88, 10126 Torino, Italy; carlo.albera@unito.it (C.A.); marina.saad@icloud.com (M.S.); paolo.solidoro@unito.it (P.S.); 7SC Medicina Interna, AO Ordine Mauriziano di Torino, Largo Turati 62, 10128 Torino, Italy; smarengo@mauriziano.it (S.M.); abriozzo@mauriziano.it (A.B.); cnorbiato@mauriziano.it (C.N.); 8Allergy and Respiratory Diseases, IRCCS Policlinico San Martino, University of Genoa, 16132 Genoa, Italy; diego.bagnasco@dimi.unige.it

**Keywords:** asthma, COPD, COVID-19, respiratory failure, respiratory support, COVID-19 outcomes, SARS-CoV-2, biologics, inhaled corticosteroids, ICS, OCS

## Abstract

*Introduction*: Asthma, along with inhaled steroids, was initially considered a risk factor for worse clinical outcomes in COVID-19. This was related to the higher morbidity observed in asthma patients during previous viral outbreaks. This retrospective study aimed at evaluating the prevalence of asthma among patients admitted due to SARS-CoV-2 infection as well as the impact of inhaled therapies on their outcomes. Furthermore, a comparison between patients with asthma, COPD and the general population was made. *Methods*: All COVID-19 inpatients were recruited between February and July 2020 from four large hospitals in Northwest Italy. Data concerning medical history, the Charlson Comorbidity Index (CCI) and the hospital stay, including length, drugs and COVID-19 complications (respiratory failure, lung involvement, and the need for respiratory support) were collected, as well as the type of discharge. *Results*: patients with asthma required high-flow oxygen therapy (33.3 vs. 14.3%, *p* = 0.001) and invasive mechanical ventilation (17.9 vs. 9.5%, *p* = 0.048) more frequently when compared to the general population, but no other difference was observed. Moreover, asthma patients were generally younger than patients with COPD (59.2 vs. 76.8 years, *p* < 0.001), they showed both a lower mortality rate (15.4 vs. 39.4%, *p* < 0.001) and a lower CCI (3.4 vs. 6.2, *p* < 0.001). Patients with asthma in regular therapy with ICS at home had significantly shorter hospital stay compared to those with no treatments (25.2 vs. 11.3 days, *p* = 0.024). *Discussion*: Our study showed that asthma is not associated with worse outcomes of COVID-19, despite the higher need for respiratory support compared with the general population, while the use of ICS allowed for a shorter hospital stay. In addition, the comparison of asthma with COPD patients confirmed the greater frailty of the latter, according to their multiple comorbidities.

## 1. Introduction

Severe acute respiratory syndrome coronavirus 2 (SARS-CoV-2) is a single-stranded RNA virus responsible for the Coronavirus Disease 2019 (COVID-19) [1]. SARS-CoV-2 clinical manifestations range from simple flu-like symptoms to extensive lung damage with respiratory failure and a high mortality rate [2].

Asthma and the use of inhaled corticosteroids (ICS) were initially considered a risk factor for worse clinical outcomes in COVID-19 [3]. This was related to the upregulation of angiotensin converting enzyme-2 (ACE-2), which represents the way that the virus enters into the cells [4,5], along with the worse clinical outcomes observed in asthma patients during previous viral outbreaks [6,7]. About 300 million people worldwide currently have asthma, which is now estimated to have a prevalence of 1–18% [8]. However, asthma prevalence seems to be increasing, thus having an extensive economic impact on national health systems. For example, the prevalence of asthma in Italy is around 4%, as approximately two million patients are currently affected [9].

Today, the prevalence of asthma in patients with COVID-19 is still debated [10,11]. Australia (21.5%), Sweden (20.2%), UK (18.2%), Netherlands (15.3%) and Brazil (13.0%) are, to date, the countries with the highest prevalence of clinical asthma. On the other hand, the lowest rates are observed in Vietnam (1.0%), Bosnia-Herzegovina (1.4%) and China (1.4%) [12].

To date, only a few studies have tried to assess the prevalence of asthma among Italian patients with COVID-19 admitted to hospital. In addition, most of them [10,11] merely focused on the prevalence of severe asthma, and the north-west area of Italy was barely included [10].

Indeed, many countries faced the very same issue, and an important step to assembling these data was obtained by the International Severe Asthma Registry, which retrospectively and prospectively collected data in patients with severe asthma [13,14]. Not only does the registry allow clinicians to estimate the prevalence of asthma in their respective countries but it also allows them to describe various asthma characteristics among their subgroups.

In addition, patients with asthma were not found to be more likely to develop severe forms of SARS-CoV-2 infections, nor COVID-19 pneumonia when compared to the general population [15,16].

On the other hand, they were observed to have more comorbidities, such as obesity [17].

The main purpose of our study is to retrospectively analyze the prevalence of asthma among patients admitted to hospital due to SARS-CoV-2 infection in Northwest Italy during the first wave pandemic, evaluate the impact of inhaled therapies on the patients’ outcomes and compare outcomes of asthma patients with those of COPD and the general population.

Moreover, we wondered whether asthma represented a risk factor for developing moderate-to-severe COVID-19 and if a higher mortality rate was observed among asthmatics.

## 2. Materials and Methods

In this retrospective study, all patients admitted for COVID-19 between February and July 2020 were included from various centers in Northwest Italy: AOU Città della Salute e della Scienza and ASO Ordine Mauriziano Hospital in Turin, AO Santa Croce e Carle in Cuneo, PO U. Parini in Aosta and AOU San Martino in Genoa.

Patients with symptoms suggestive of COVID-19 but with a negative rhino-pharyngeal swab, and patients hospitalized for other causes—with a positive swab during hospital stay but no symptoms attributable to COVID-19—were excluded.

The following data were collected from all patients: demographics (age, sex), smoking habits, history of asthma and/or COPD, comorbidities, home therapy for asthma or COPD, course of hospitalization for COVID-19 and outcome.

### 2.1. Diagnosis of COVID-19

COVID-19 diagnosis was made according to the WHO recommendations [18], based on the finding of one or more nasopharyngeal molecular swabs (MDX DiaSorin, Cypress, San Jose, CA, USA and Ingenius Elitech, Puteaux, France) and COVID-19 suggestive symptoms (fatigue, general malaise, headache, loss of taste and/or smell, fever, rhinitis, cough, dyspnea and myalgia).

### 2.2. COPD and Asthma Diagnosis

All patients who did a pneumological or allergy examination with a diagnosis of asthma and performed at least one spirometry compatible with asthma [8] in the 6 months prior to admission were classified as asthmatics.

All patients who had undergone a pneumological visit with a diagnosis of COPD and performed at least one COPD-compatible spirometry [19] in the six months prior to admission were classified as having COPD.

Concerning patients with an obstructive pathology, the related inhalation or home systemic therapies were also analyzed (SABA, LABA, ICS, LTRA, LAMA, OCS, biologics, etc.) as well as long-term oxygen therapy (LTOT).

### 2.3. Comorbidity

The presence of comorbidities was evaluated by using the Charlson Comorbidity Index (CCI); CCI is a prognostic model able to predict one-year mortality risk based on the individual severity of comorbidities. The information to fill the CCI in were taken from the diagnostic codes obtained from the hospital folders. As patients with obstructive lung diseases were included, a CCI adjusted for asthma was used [20].

### 2.4. Evaluation of the Hospital Stay

To assess the progress of the disease during hospitalization, the following parameters were collected and analyzed: Length of hospital stay, calculated from the day of admission for signs or symptoms related to SARS-CoV-2 to the day of discharge or demise; the presence of respiratory failure [21,22], evaluated by measuring the arterial P/F ratio; the presence of a COVID-19 pneumonia [23,24,25], assessed by chest HRTC; the need for ventilatory support (O2 in nasal cannula (NC), high flow nasal cannula (HFNO), Non-invasive Ventilation (NIV), Invasive Mechanical Ventilation (IMV)); therapies carried out during hospitalization, including systemic corticosteroids, hydroxychloroquine, monoclonal antibodies (tocilizumab, baricitinib) and antiviral drugs (Remdesivir, Lopinavir or Ritonavir) [26]; type of discharge: discharge (at home or in the facility) or death.

### 2.5. Statistical Analysis

IBM SPSS Statistics for Windows, version 26 (IBM Corp., Armonk, NY, USA) was used to perform statistical analysis. The Kolmogorov–Smirnov normality test was chosen to assess the normality distribution of data, followed by a descriptive analysis of the variables.

Baseline characteristics were evaluated and expressed as mean (± standard deviation, SD) unless otherwise specified, for continuous variables; absolute and relative frequencies are shown for categorical variables.

The Student *t*-test and Chi-square test were performed to compare baseline characteristics of patients with asthma vs. obstructive lung disease and COPD.

A *p*-value below 0.05 was considered statistically significant.

### 2.6. Ethical Committee

The study was approved by the ethics committee “Comitato etico interaziendale AOU Città della Salute e della Scienza di Torino—AO Ordine Mauriziano di Torino—ASL Città di Torino”, case file no. 00356/2020, approved on September 15th 2020, protocol no. 0086037, and it was conducted according to the Helsinki Declaration.

As per derogations established by the ethics committee, under article 110 d.lgs 196/2003 paragraph 2, modified in 2018, “Consent shall not be necessary if, under specific circumstances, it is impossible or unreasonably difficult to inform data subjects or such events might prevail or affect the main purpose of the study. In these circumstances, the data controller takes proper steps to protect the rights, the freedom and the legitimate interests of those who are involved, and the study is adequately justified and approved by the geographically competent ethics committee as well as being authorized by the Garante also in pursuance of Article 36”; given the retrospective nature of the study and in consideration of those who cannot be informed (e.g., patients who passed away), informed consent was not obtained from all the patients.

## 3. Results

One thousand and sixteen patients (N = 1016) were recruited in the study. Among these, 906 didn’t have any history of obstructive lung disease (controls), 71 (7.0%) suffered from COPD and 39 (3.8%) patients had a diagnosis of asthma. Demographic data and comorbidities of the enrolled sample are listed in Table 1.

All the patients with obstructive lung disease received a proper maintenance prescription for their disease. However, among patients with asthma, 12 subjects (30.8%) did not take any medication and 5 (12.8%) only took SABA as needed; twenty-two patients (56.4%) regularly used ICS, alone or in association with another controller, while 3 (7.7%) required chronic OCS and 2 patients (5.1%) were administered with biologics, respectively. Among patients with COPD, 3 (4.2%) used to take SABA alone, 16 (22.5%) were in regular ICS treatment, one (1.4%) was in chronic OCS and 13 (18.3%) did not take any home treatment. Eight patients (11.3%) were at home with LTOT. Home medications of patients with COPD are shown in Figure 1.

### 3.1. Comparison between Patients with Asthma and Controls

Patients with asthma showed a lower median age compared with controls (59.2 [14–96] vs. 64.7 14–96] years, *p* = 0.037), but no difference was found between the groups for sex distribution, smoking habits or comorbidities (Table 1).

In terms of clinical trends and outcomes, patients with asthma showed significantly lower prevalence of COVID-19 pneumonia and more frequently required HFNO therapy as well as invasive mechanical ventilation (33.3% vs. 14.3%, *p* = 0.001 and 17.9% vs. 9.5%, *p* = 0.048 respectively).

No difference was found concerning the average length of hospital stay nor for the frequency of respiratory failure, COVID-19 treatments or mortality rate (Table 2).

### 3.2. Comparison between Patients with Asthma and COPD

Patients with asthma, compared with COPD, had a greater prevalence of females (51.3% vs. 28.2%) and were younger (59.2 [14–96] vs. 76.8 [42–94] years, *p* < 0.001) (Table 1). Asthma patients were more often non-smokers (Table 2) and had a lower Charlson Comorbidity Index (3.4 ± 1.9 vs. 6.2 ± 2.5, *p* < 0.001).

As regards hospitalization, patients with asthma had a shorter length of hospital stay, although the difference was not significant. Similarly, COVID-19 pneumonia was observed less frequently among patients with asthma compared with COPD (53.8% vs. 66.2%, *p* = 0.022).

Concerning the need for respiratory support, patients with asthma received NIV less frequently (15.4% vs. 36.6%, *p* = 0.019), but, on the other hand, invasive mechanical ventilation was administered more often when compared to patients with COPD (17.9% vs. 9.9%, *p* = 0.044) (Table 2).

A lower mortality rate was observed in patients with asthma compared with COPD (15.4% vs. 39.4%, *p* < 0.001) but no difference in the prevalence of respiratory failure or administered COVID-19 treatments was observed.

### 3.3. Comparison between Asthma Patients without Medications and in Regular ICS

Patients with no regular treatment at home (SABA as needed or no treatment at all) showed a significantly longer hospital stay compared to those in regular treatment with ICS (25.2± 15.1 vs. 11.4 ± 9.9 days, *p* = 0.024). However, no difference was observed among the other analyzed variables (Respiratory failure, COVID-19 pneumonia, Respiratory support, Mortality risk), data not shown.

## 4. Discussion

As per our knowledge, this is the very first study aimed at evaluating the prevalence of asthma among patients admitted due to SARS-CoV-2 infection in Northwest Italy; in addition, the key features of our study were the comparisons between patients with asthma, COPD and the general population.

Asthma currently affects about 4% of the whole Italian population [27], with a great variability between different regions [27]. In our cohort, the prevalence of asthma was observed to be 3.83%, thus providing further proof that patients with asthma did not fall sick more frequently than the general population.

This might be explained by analyzing the behavior of asthma patients in the first wave of COVID-19, as they took a more cautious approach with a strict usage of personal protective equipment (PPE).

In terms of the outcome of viral infections, patients with asthma were found to have a worse trend during the previous epidemics by coronaviruses [28].

In addition, our study proved there was no statistical difference in the mortality rate between hospitalized asthma patients and the general population, in agreement with other research which showed the risk of getting infected by SARS-CoV-2 was not increased [10,11,29]. Similarly, the outcome for COVID-19 did not appear to be worse [10,11].

In our cohort, only two patients were in therapy with biological drugs, suggesting that most of them had a mild-to-moderate asthma which was likely to be kept under control with a proper adherence to standard therapies [30,31].

However, asthma patients required HFNO and IMV more frequently than the control group; this element potentially suggested a more severe infection, even though, to date, no plausible explanation has been given. Such differences might be explained by airway remodelling observed in patients who have been suffering from asthma for a long time, in mild forms or in patients with a low therapeutic compliance [32].

All the patients enrolled in our study had, in fact, a proper prescription as per GINA guidelines [8]; however, only half of them took their prescription regularly. Adherence to medication regimes with ICS in asthma has always been notoriously low, with estimates showing as little as 30% [30]. Nevertheless, in the first wave of the pandemic, such a percentage increased by 14% therefore varying between 54 and 61% [33], mainly due to the fear of worse clinical outcomes, as previously stated.

Many studies managed to underline the protective mechanism of ICS in patients with asthma or COPD; it has been shown that SARS-CoV-2 requires ACE2 as its entry receptor.

However, whether or not up- or downregulation of ACE2 expression in sputum, airways or lungs has clinical effects on virulence or outcomes of COVID-19 still needs to be clarified. ACE2 is downregulated in asthma, and this effect may be due to the suppressive effects of ICS use on the replication of SARS-CoV-2 [34].

Despite such elements, no statistically significant difference was observed in ventilation, onset of complications related to COVID-19 or mortality rates between asthma patients in chronic therapy with ICS and those who did not take it regularly; however, hospital stay was found to be longer for the latter. This element indeed proves to be interesting as it is well established that patients with asthma have heightened airway hyperresponsiveness (AHR) for various stimuli [35], allergens and pathogens [36]. Despite the SABA-induced rapid effect on relieving symptoms related to bronchoconstriction, no difference on airway inflammation nor on viral replication can be observed [37]. Moreover, a more frequent usage of SABA is related to an increased risk of mortality for asthma [11,38] as it is burdened with many adverse effects (e.g., tachycardia, tremors, anxiety) which might have contributed to lengthening the hospital stay of these patients [39].

In addition, the comparison of asthma with COPD patients confirmed the greater frailty of the latter, as they were characterized by both higher CCI and increased mortality rate. Such findings are consistent with other studies [40,41], which underlines the increased risks of negative outcomes in patients with COPD and COVID-19. Indeed, these patients received NIV more often, but invasive mechanical ventilation less frequently. This finding might be explained by analyzing the clinical pathway currently in use for the management of patients with COPD; intubation is usually reserved for patients in which other forms of ventilation are not sufficient nor feasible, since any oxygen administration is hard to wean or reduce when compared to the general population or asthma patients [42].

## 5. Conclusions

Our analysis suggests that asthma does not confer an increased risk of severe complications related to COVID-19 when compared to the general population. The role of ICS in preventing severe forms of COVID-19 is currently under evaluation, but more research is needed. No statistical difference in the mortality rate between hospitalized asthma patients and the general population was observed, in line with other similar studies, despite the fact that asthma patients required HFNO and IMV more frequently than the control group. On the other hand, patients with COPD showed overall greater frailty, thus leading to a worse outcome of covid-19 as compared to the general population and the asthmatics as well.

## 6. Limitations of the Study

This study has potential limitations to consider. The results in the model are based on retrospective data, gathered from medical records and clinical documentation. Furthermore, patients were only enrolled from large centers rather than COVID-only hospitals; therefore, a selection bias might be present. The very same issue might apply to the relatively small number of patients with asthma in this study as they were either more cautious (especially in the first wave of this pandemic) or quite scared to require medical assistance in ER. Moreover, data on pre-admission therapies for asthma (such as the percentage of COPD patients that were on LAMA, LAMA/LABA combination or even triple therapy) were limited, thereby preventing the exact knowledge of the severity of each patient or how controlled was their asthma before COVID-19.

## Figures and Tables

**Figure 1 jpm-12-01184-f001:**
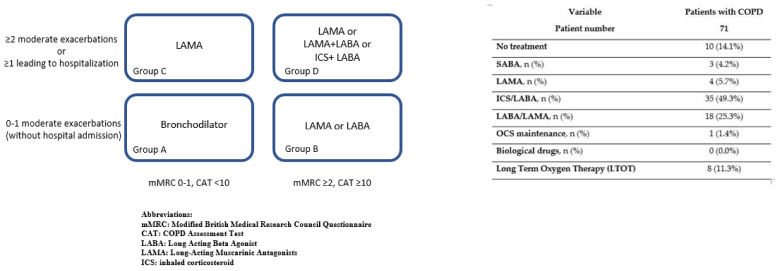
Home treatment in patients affected with COPD.

**Table 1 jpm-12-01184-t001:** Demographic data and comorbidities in patients with asthma compared with COPD or no history of obstructive lung disease.

	Patients with Asthma	Patients with COPD	Asthma vs. COPD	Patients with No Obstructive Lung Disease (Controls)	Asthma vs. Controls
	39	71	*p*	906	*p*
**Females**, *n* (%)	20 (51.3%)	20 (28.2%)	0.016	354 (39.1%)	n.s.
**Age, years** (mean [range])	59.2 [14–96]	76.8 [42–94]	<0.001	64.7 [14–96]	0.037
**Charlson Comorbidity Index (CCI)** (mean ± SD)	3.4 ± 1.9	6.2 ± 2.5	<0.001	3.5 ± 2.6	n.s.
**Smoking habits**					
Non-smokers, n (%)	25 (64.1%)	6 (8.5%)	<0.001	417 (46.0%)	n.s.
Smokers, n (%)	3 (7.7%)	13 (18.3%)	<0.001	157 (17.3%)	n.s.
Former smokers, n (%)	6 (15.4%)	50 (70.4%)	<0.001	282 (31.1%)	n.s.
Not declared, n (%)	5 (12.8%)	2 (2.8%)	<0.001	50 (5.6%)	n.s.

**Table 2 jpm-12-01184-t002:** Clinical trend and outcomes in patients with asthma compared to COPD or no history of obstructive lung disease.

Variable	Patients with Asthma	Patients with COPD	Asthma vs. COPD	Patients with No Obstructive Lung Disease (Controls)	Asthma vs. Controls
Patient Number	39	71	*p*	906	*p*
**Length of hospital stay, days (mean ± SD)**	17.4 ± 19.2	24.5 ± 23.7	n.s.	18.1 ± 15.8	n.s.
**Complications**					
Respiratory failure, n (%)	28 (71.8%)	59 (83.1%)	n.s.	641 (70.8%)	n.s.
COVID-19 pneumonia, n (%)	21 (53.8%)	47 (66.2%)	**0.022**	622 (68.7%)	0.005
**Ventilation**					
No oxygen supports, n (%)	5 (12.8%)	4 (5.6%)	n.s.	216 (23.8%)	n.s.
Nasal cannula (NC), n (%)	8 (20.5%)	14 (19.7%)	n.s.	237 (26.2%)	n.s.
High-flow nasal oxygen (HFNO), n (%)	13 (33.3.%)	20 (28.2%)	n.s.	130 (14.3%)	**0.001**
Non-invasive Ventilation (NIV), n (%)	6 (15.4%)	26 (36.6%)	**0.019**	237 (26.2%)	n.s.
Invasive Mechanical Ventilation (IMV), n (%)	7 (17.9%)	7 (9.9%)	**0.044**	86 (9.5%)	0.048
**Hospital treatment**					
Inhaled corticosteroids, n (%)	12 (30.8%)	18 (25.4%)	n.s.	6 (0.7%)	**<0.001**
Hydroxychloroquine, n (%)	23 (59.0%)	44 (62%)	n.s.	657 (72.3%)	n.s.
Antiviral drugs, n (%)	18 (46.2%)	36 (50.7%)	n.s.	541 (59.5%)	n.s.
Monoclonal antibodies, n (%)	5 (12.8%)	6 (8.5%)	n.s.	161 (17.7%)	n.s.
**Deaths, n (%)**	6 (15.4%)	28 (39.4%)	**<0.001**	156 (17.2%)	n.s.

## Data Availability

Not applicable.

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
