# Peer review of "COPD, but Not Asthma, Is Associated with Worse Outcomes in COVID-19: Real-Life Data from Four Main Centers in Northwest Italy"

_jpm, 2022, doi:10.3390/jpm12071184_

Round 1

Reviewer 1 Report

An interesting topic to analyse and write on as globally, there is a trend where asthmatics have milder forms of COVID-19 associated complications eg pneumonia as compared to other comorbidities. Please include these efforts into your introduction and cite relevant studies/ observations that have been published. Rather than focusing on asthma in Northwest Italy, the authors need to generalise their introduction, results and discussion to make it applicable to the rest of the world. This is the biggest limitation of the paper. 

It is surprising to see only a small percentage of population with asthma / COPD out of the 1016 patients that were recruited. Was this due to selection bias or is this in keeping with the prevalence of asthma / COPD in the general population? 

Under results, a suggestion is to table the medications that the patients are on which would make it more easily readable. How many percentage of COPD patients were on LAMA, LAMA/LABA combination or even triple therapy? 

The discussion section would need to be elaborated. The biggest limitation is the small number of asthmatic / COPD patients in this study, making it difficult to draw any general conclusions from the study and this should be acknowledged. 

Moreover, if the results show that asthmatic patients tend not to develop COVID-19 pneumonia, why was there such a need for HFNO and IV compared to the general population? 

The article does not have a conclusion that can be generalised to the entire world. 

Author Response

My colleagues and I are extremely grateful for the reviewers' comments.

Please find attached a point-by-point reply.

Reviewer 2 Report

In this paper, it is analyzed the differences in the course and outcome of treatment of COVID-19 infection between hospitalized patients with asthma, chronic obstructive pulmonary disease and the control group of patients who did not suffer from obstructive pulmonary disease. 

Several interesting and significant results were obtained, of which perhaps the most significant was that patients with asthma had a significantly lower incidence of pneumonia caused by Covid-19 infection compared to patients with COPD and the control group, as well as lower mortality compared with patients with COPD.

Introduction - the research issues are clearly presented, it is pointed out that a limited number of studies dealing with the mentioned research problem are available in the literature (although a number of papers dealing with the context of asthma and Covid-19 have been published so far, it is possible that at the time of writing, there were not too many available sources, as this research covered the first months of the pandemic), the objectives of the research at the end of the paper were clearly set;
Methodology - extremely clearly and precisely written, without objections;
Results - the results of the research are very clearly and simply presented, textually and / or tabularly. The data in the tables are for the most part clearly presented, with the proviso that below all the tables there is a legend with an interpretation of the abbreviations used;
Discussion - in this chapter the obtained research results are commented. In some parts of the text, these statements should be supported by appropriate references (eg the last sentence of the second paragraph), while the parts of the text in which the authors hypothesized possible explanations for their results are clearly marked. Given the results of this study, as well as the previously shown positive effect of inhaled corticosteroids in the treatment of Covid-19 infection (which also belong to the group of basic drugs in the treatment of asthma), I believe that this paper would have greater value if the authors treated more this aspect of the researched issue. This could be done in the form of a brief literature review that will analyze studies that have addressed the possible (protective) effect of inhaled corticosteroids in asthmatics on the development and course of Covid-19 infection, if such studies are available.
Conclusion - Although the Conclusion chapter in the Instructions for Authors is optional, I believe that a short conclusion summarizing the most significant research results would stylistically round out and enrich the manuscript.
References - references should be given in the manner specified in the Guidelines for Authors.

I believe that the English language should generally be improved, especially in the abstract and introduction of the manuscript, and that some long sentences should be shortened, so that the text would be more understandable and easier to read. Typographical errors can be noticed in several places in the text (in the discussion chapter, paragraph 5 - there is an error in quoting the reference, paragraph 7 - point before reference no. 4).

Overall, the paper is interesting, readable, with clearly presented and significant results that would contribute to an increase in the total amount of knowledge about Covid-19 disease, which is why I am of the opinion that it should be accepted after the stated (minor) corrections.

Author Response

My colleagues and I are extremely grateful for the reviewers'comments.

Please, find attached the point-by-point reply.

Round 2

Reviewer 1 Report

The authors have addressed the concerns and suggestions adequately. No further changes are necessary.